Thalassemys bruntrutana n. sp., a new coastal marine turtle from the Late Jurassic of Porrentruy (Switzerland), and the paleobiogeography of the Thalassemydidae

Püntener Christian christian.puntener@jura.ch
Anquetin Jérémy
Billon-Bruyat Jean-Paul
Section d’archéologie et paléontologie, Office de la Culture, République et Canton du Jura , Porrentruy , Switzerland
Wedel Mathew
Electronic publication date: 2015 Sep 29
Publication date: 2015
Volume: 3
Electronic Location ID: e1282
Received 2015 Jul 22; Accepted 2015 Sep 8
Copyright: © 2015 Püntener et al.
Copyright year: 2015
Copyright holder: Püntener et al.
License: This is an open access article distributed under the terms of the Creative Commons Attribution License, which permits unrestricted use, distribution, reproduction and adaptation in any medium and for any purpose provided that it is properly attributed. For attribution, the original author(s), title, publication source (PeerJ) and either DOI or URL of the article must be cited.
License URL: https://creativecommons.org/licenses/by/4.0/

Keywords: Thalassemys bruntrutana, Thalassemydidae, Testudines, Kimmeridgian, Switzerland

Funding: Federal Roads Office (FEDRO, 95%) Republic and Canton of Jura (RCJU, 5%) The PAL A16 team (Section d’archéologie et paléontologie) is funded by the Federal Roads Office (FEDRO, 95%) and the Republic and Canton of Jura (RCJU, 5%). The funders had no role in study design, data collection and analysis, decision to publish, or preparation of the manuscript.

==============================
Background. The Swiss Jura Mountains are a key region for Late Jurassic eucryptodiran turtles. Already in the mid 19th century, the Solothurn Turtle Limestone (Solothurn, NW Switzerland) yielded a great amount of Kimmeridgian turtles that are traditionally referred to Plesiochelyidae, Thalassemydidae, and Eurysternidae. In the past few years, fossils of these coastal marine turtles were also abundantly discovered in the Kimmeridgian of the Porrentruy region (NW Switzerland). These findings include numerous sub-complete shells, out of which we present two new specimens of Thalassemys (Thalassemydidae) in this study.

Methods. We compare the new material from Porrentruy to the type species Th. hugii, which is based on a well preserved specimen from the Solothurn Turtle Limestone (Solothurn, Switzerland). In order to improve our understanding of the paleogeographic distribution of Thalassemys, anatomical comparisons are extended to Thalassemys remains from other European countries, notably Germany and England.

Results. While one of the two Thalassemys specimens from Porrentruy can be attributed to Th. hugii, the other specimen represents a new species, Th. bruntrutana n. sp. It differs from Th. hugii by several features: more elongated nuchal that strongly thickens anterolaterally; wider vertebral scales; proportionally longer plastron; broader and less inclined xiphiplastron; wider angle between scapular process and acromion process. Our results show that Th. hugii and Th. bruntrutana also occur simultaneously in the Kimmeridgian of Solothurn as well as in the Kimmeridgian of England (Kimmeridge Clay). This study is an important step towards a better understanding of the paleobiogeographic distribution of Late Jurassic turtles in Europe.

Introduction

Thalassemys Rütimeyer, 1873 is a coastal marine turtle from the Late Jurassic of Western Europe (Bräm, 1965; Lapparent de Broin F de, 2001). It is the only currently recognized representative of the family Thalassemydidae Zittel, 1889, a group that is potentially related to the Plesiochelyidae Baur, 1888 and the Eurysternidae Dollo, 1886 (Joyce, 2007). However, the exact relationships and systematics of these three groups are rather confused (e.g., Broin, 1994; Lapparent de Broin, Lange-Badré & Dutrieux, 1996; Joyce, 2003; Joyce, 2007; Anquetin & Joyce, 2014).

Rütimeyer (1859a) and Rütimeyer (1859b) first mentioned the name Thalassemys during a conference of the Schweizerische Naturforschende Gesellschaft at the University of Bern (Maack, 1869), but the name only became available in 1873 when he described and figured two species of Thalassemys from the Kimmeridgian of Solothurn, Canton of Solothurn, Switzerland: Th. hugii Rütimeyer, 1873 and Th. gresslyi Rütimeyer, 1873. The type species Th. hugii is based on a relatively flat shell with associated postcranial remains (NMS 8595–8609; formerly NMS 1), which is the largest turtle in Solothurn (preserved carapace length of 630 mm). Since Rütimeyer (1873), the fossil turtles of the so-called Solothurn Turtle Limestone have undergone two major revisions (Bräm, 1965; Anquetin, Püntener & Billon-Bruyat, 2014), both with important impacts on the taxonomy of Thalassemys. Bräm (1965) synonymized Th. gresslyi with Th. hugii and erected  a new species, Th. moseri Bräm, 1965. Revealing the presence of lateral plastral fontanelles in Thalassemys, Anquetin, Püntener & Billon-Bruyat (2014) synonymized Eurysternum ignoratum Bräm, 1965 with Th. hugii and excluded ‘Th.’ moseri from Thalassemys.

There is some discussion regarding the validity and taxonomy of ‘Th.’ moseri in the literature (Lapparent de Broin, Lange-Badré & Dutrieux, 1996; Anquetin, Püntener & Billon-Bruyat, 2014; Pérez-García, 2015), but all authors agree that this taxon should be excluded from Thalassemys. This species was described by Bräm (1965) based on two shells from the Kimmeridgian of Solothurn, Switzerland. A specimen from the Tithonian of western France consisting of an association between a skull and a partial shell was later referred to this taxon (Rieppel, 1980). Without revising the material, Lapparent de Broin, Lange-Badré & Dutrieux (1996) stated that ‘Th.’ moseri was invalid since they referred the two specimens from Solothurn (type material) to juvenile forms of Plesiochelys solodurensis Rütimeyer, 1873, a plesiochelyid from the same locality. However, they considered the material described by Rieppel (1980) as a distinct and indeterminate species of Plesiochelys Rütimeyer, 1873. In 2014, Anquetin, Püntener & Billon-Bruyat (2014) revised the type material from Solothurn and concluded that ‘Th.’ moseri was indeed a valid taxon, distinct from Plesiochelys solodurensis. In the meantime, Pérez-García (2015) built upon the conclusions of Lapparent de Broin, Lange-Badré & Dutrieux (1996) and referred the material described by Rieppel (1980) to a new taxon, Jurassichelon oleronensis, without reassessing the validity of ‘Th.’ moseri. It should be noted that the latter author failed to see both the material from Solothurn and the specimen described by Rieppel (1980). Sorting out this situation would require a complete revision of the material, something that has not been done since 1980. It should notably be determined whether the material described by Rieppel (1980) and the material from Solothurn belong to the same species. This revision is ongoing, and the present paper on Thalassemys is certainly not the place to discuss further the systematics of this unrelated taxon.

Several other species have been attributed to Thalassemys in the late 18th and early 19th centuries, including Thalassemys marina Fraas, 1903 (Tithonian of Schnaitheim, Baden-Württemberg, Germany), Thalassemys heusseri Oertel, 1924 (Tithonian of Holzen im Hils, Lower Saxony, Germany), and Thalassemys ruetimeyeri Lydekker, 1889 (Berriasian of Dorset, England). Because the type material is lost and the original description is insufficient, Th. heusseri must be considered a nomen dubium. Thalassemys ruetimeyeri has been recognized as a junior synonym of the pleurosternid Dorsetochelys typocardium (Seeley, 1869) (Milner, 2004; Pérez-García, 2014). Thalassemys marina has long been considered to represent a eurysternid (see Discussion), but Anquetin, Püntener & Billon-Bruyat (2014) recently challenged this conclusion and confirmed the validity of this taxon. As a result, Th. hugii and Th. marina are currently considered as the only two valid thalassemydids. Undetermined Thalassemys remains have been recently reported from the Kimmeridgian near Oker, Lower Saxony, Germany (Marinheiro & Mateus, 2011), and from the Kimmeridgian of the Isle of Purbeck, Dorset, southern England (Pérez-García, 2015).

In the present study we describe two new specimens of Thalassemys (MJSN SCR011-87 and MJSN BSY008-905) from the upper Kimmeridgian of Porrentruy, Canton of Jura, Switzerland. They were recently discovered within the scope of the Paleontology A16 project, which aims to rescue the paleontological material found during the construction of the A16 Transjurane highway. The excavations resulted in a rich and diverse vertebrate fossil collection from the Kimmeridgian, notably including extensive dinosaur trackways (Marty & Hug, 2003; Marty et al., 2007; Marty, 2008; Marty & Billon-Bruyat, 2009) and numerous coastal marine turtles (Billon-Bruyat, 2005; Püntener et al., 2014; Anquetin, Püntener & Billon-Bruyat, 2015). MJSN SCR011-87, an articulated, sub-complete shell with an associated scapula, is referred to a new species, Thalassemys bruntrutana n. sp. In contrast, MJSN BSY008-905, which consists of disarticulated shell elements and a partial scapula, is referred to the type species Th. hugii. Furthermore, we discuss the implications of the new material from Porrentruy for the taxonomy and paleobiogeography of Thalassemys.

Geological Setting

The new Thalassemys specimens were collected near the village of Courtedoux, along the A16 Transjurane highway in the Ajoie Region, Canton of Jura, NW Switzerland (Fig. 1). MJSN SCR011-87 was discovered in Sur Combe Ronde (SCR) in 2011, and MJSN BSY008-905 in Bois de Sylleux (BSY) in 2008. Both specimens come from the Lower Virgula Marls (Reuchenette Formation, Chevenez Member) that are dated from the Eudoxus ammonite zone (Comment et al., in press) (Fig. 2). The Lower Virgula Marls are slightly older than the Solothurn Turtle Limestone, which forms the uppermost member of the Reuchenette Formation and is dated from the Autissiodorensis ammonite zone (Meyer, 1994; Comment, Ayer & Becker, 2011).

Figure 1 Geographical map of the Ajoie Region, Canton Jura, Switzerland.

The excavation sites Sur Combe Ronde (SCR) and Bois de Sylleux (BSY) are situated along the Transjurane A16 highway (gray; dotted lines indicate tunnels).

Figure 2 Stratigraphic section of the Reuchenette Formation.

The two Thalassemys specimens were discovered within the Lower Virgula Marls (Eudoxus ammonite zone). Scheme modified after Comment et al. (in press).

MJSN SCR011-87 was embedded in a hardground within the Lower Virgula Marls (Fig. 2). The reddish and strongly encrusted (Ostreidae, Serpulidae) hardground is dated from the Orthocera ammonite horizon (Comment et al., in press). Its invertebrate fauna includes brachiopods (Sellithyris) and benthic bivalves (Ceratomya, Pholadomya), the latter being responsible for the hardgrounds undulated surface. Apart from this sub-complete shell, vertebrates are limited to isolated material (chondrichthyans, osteichthyans, turtles, crocodilians) and two well preserved crocodilian skeletons (Metriorhynchus sp. and Steneosaurus cf. bouchardi; Schaefer, 2012; Schaefer & Billon-Bruyat, 2014).

MJSN BSY008-905 comes from a marly interval of the Lower Virgula Marls that is slightly younger (Hibridus ammonite horizon) than the aforementioned hardground (Koppka, 2015; Comment et al., in press) (Fig. 2). This about 1 m-thick brown marl yielded a rich and diverse coastal marine assemblage, including invertebrates (bivalves, gastropods, cephalopods, crustaceans, and echinoderms), vertebrates (chondrichthyans, osteichthyans, turtles, crocodilians, and pterosaurs), and wood remains Billon-Bruyat, 2005; Marty & Billon-Bruyat, 2009; Philippe et al., 2010; Koppka, 2015).

Nomenclatural Acts

The electronic version of this article in Portable Document Format (PDF) will represent a published work according to the International Commission on Zoological Nomenclature (ICZN), and hence the new names contained in the electronic version are effectively published under that Code from the electronic edition alone. This published work and the nomenclatural acts it contains have been registered in ZooBank, the online registration system for the ICZN. The ZooBank LSIDs (Life Science Identifiers) can be resolved and the associated information viewed through any standard web browser by appending the LSID to the prefix “http://zoobank.org/”. The LSID for this publication is: urn:lsid:zoobank.org:pub:5206D70B-E07B-42A0-AF98-E0371DD491D5. The online version of this work is archived and available from the following digital repositories: PeerJ, PubMed Central and CLOCKSS.

Systematic Paleontology

TESTUDINES Batsch, 1788	
PANCRYPTODIRA Joyce, Parham & Gauthier, 2004	
THALASSEMYDIDAE Zittel, 1889	

Remark. Thalassemydidae is currently a monogeneric family. Since 2014, we are engaged in a global revision of European Late Jurassic coastal marine turtles (this represents more than 60 named species in the literature), which are traditionally referred to three families: Plesiochelyidae, Eurysternidae and Thalassemydidae. The characteristics and potential relationships of these three families are still obscure. Early on, we decided to take a conservative path and keep on using these three family names in their traditional definition until we have gathered enough knowledge about their phylogenetic relationships (Anquetin, Püntener & Billon-Bruyat, 2014). The purpose was to avoid havocking supra-generic classification at each new paper, especially without phylogenetic justification.

Thalassemys Rütimeyer, 1873

Type species. Thalassemys hugii Rütimeyer, 1873

Included valid species. Thalassemys hugii Rütimeyer, 1873; Thalassemys marina Fraas, 1903; Thalassemys bruntrutana n. sp.

Occurrence. Kimmeridgian of Switzerland (Rütimeyer, 1873; Bräm, 1965; this study), Germany (Marinheiro & Mateus, 2011), and England (Pérez-García, 2015; Pérez-García, in press); Tithonian of Germany (Fraas, 1903).

Revised diagnosis. Type and only genus of Thalassemydidae. Medium to large sized turtle (estimated carapace length up to 700 mm); relatively flat carapace (as opposed to the more domed plesiochelyid carapace); presence of clearly visible linear striations perpendicular to most sutures on the carapace and plastron. Differing from Plesiochelys, Craspedochelys Rütimeyer, 1873, and Tropidemys Rütimeyer, 1873 in: presence of small costo-peripheral fontanelles in the adult; presence of lateral plastral fontanelles; non-sutural connection of epi- and entoplastron; presence of a small fontanelle between the xiphiplastra; wider angle between scapular process and acromion process (only known in Plesiochelys). Differing from Idiochelys Meyer, 1839a, Eurysternum Meyer, 1839b, and Solnhofia Gaffney, 1975 in: larger size; narrower vertebral scales; osseous bridge; complete series of neurals (incomplete in Idiochelys); central plastral fontanelle present (absent in Idiochelys) and longer than wide (opposite in Eurysternum); small xiphiplastral fontanelle present (absent in Idiochelys and Solnhofia).

Remarks. Thalassemys differs from other Late Jurassic turtles from Europe by a combination of several features. A striking character is the presence of distinct linear striations perpendicular to sutures (Anquetin, Püntener & Billon-Bruyat, 2014). This somewhat recalls the condition known in the Early Cretaceous Pleurosternon bullockii Owen, 1842 (see Milner, 2004), but these striations are more pronounced in Thalassemys. In contrast, they are absent (or only very weakly expressed) in plesiochelyids and eurysternids. The shape of vertebral scales is also characteristic of Thalassemys. The anterolateral sides of vertebrals 2–4 are slightly concave, whereas the posterolateral sides are either slightly convex or sub-straight. The plastral anatomy of Thalassemys clearly differentiates it from plesiochelyids and eurysternids. In contrast to plesiochelyids, lateral plastral fontanelles occur in Thalassemys and there is no sutural connection of the epiplastra and entoplastron with the hyoplastra. In contrast to eurysternids, the bridge of Thalassemys is osseous. As previously proposed by Bräm (1965), the angle between the scapular and acromion processes of the scapula may also be a distinguishing feature of Thalassemys. This angle is more open in Thalassemys (113–130°) than in Plesiochelys etalloni (Pictet & Humbert, 1857) (103–105°), but a broader survey of Late Jurassic coastal marine turtles is needed to definitely conclude on this character variation (Table 1).

Table 1 The angle between scapular process and acromion process in specimens of Thalassemys and Plesiochelys.

When the two scapulae were preserved, both were measured. The high value for the right scapula of NMS 8631–8643 must therefore be treated with caution with regard to taphonomic compaction.

Specimen	Species	Scapular angle	
NMS 8595–8609	Th. hugii	118°	
NMS 8612–8627	Th. hugii	114°/113°	
NMS 8631–8643	Th. hugii	113°/122°	
MJSN BSY008-905	Th. hugii	116°	
MJSN SCR011-87	Th. bruntrutana	130°	
NHMUK R8699	Th. bruntrutana a	103°	
NMS 8584	P. etalloni	103°	
NMS 8731	P. etalloni	103°	
NMS 9153	P. etalloni	105°	
Notes.

a Thalassemys sp. in Pérez-García (2015).

Anquetin, Püntener & Billon-Bruyat (2014) suggested that a strong anterior widening of the first neural was diagnostic for Thalassemys. This feature is present in most specimens referred to Th. hugii, as well as in the type specimens of Th. marina and Th. bruntrutana. However, Thalassemys specimens from England (NHMUK R8699 and OUMNH J.66966; see Discussion) and one specimen of Th. hugii from Solothurn (NMS 8555) lack a strong anterior widening of the first neural. This feature is therefore probably variable intraspecifically.

Thalassemys hugii Rütimeyer, 1873

Synonymy. Thalassemys Gresslyi Rütimeyer, 1873 and Eurysternum ignoratum Bräm, 1965 (Anquetin, Püntener & Billon-Bruyat, 2014).

Type material. NMS 8595–8609, almost complete and articulated carapace, disarticulated plastral fragments, postcranial remains. Lectotype designated by Bräm (1965: 143).

Illustrations of type. Rütimeyer (1873: plate I); Bräm (1965: plate 7); Anquetin, Püntener & Billon-Bruyat (2014: Figs. 6A–6D); Figs. 3C–3D, 6C–6D and 7C–7D.

Figure 3 Carapaces of Thalassemys.

(A, B) Thalassemys bruntrutana, specimen MJSN SCR011-87 (Kimmeridgian, Porrentruy, Switzerland); (C, D) Thalassemys hugii, specimen NMS 8595–8609 (Kimmeridgian, Solothurn, Switzerland). Line width indicates natural borders (thick lines), bone sutures (medium lines), and fractures (thin lines); double lines indicate scale sulci; matrix is gray. Abbreviations: co, costal; n, neural; sp, suprapygal; v, vertebral scale; *, intermediate element (see text).

Type horizon and locality. Solothurn Turtle Limestone, uppermost member of the Reuchenette Formation (Autissiodorensis ammonite zone, upper Kimmeridgian, Late Jurassic), vicinity of Solothurn, Canton of Solothurn, Switzerland.

Occurrence. Kimmeridgian of Switzerland (Solothurn and Porrentruy) and England (Abingdon).

Referred specimens. All specimens referred by Bräm (1965) and Anquetin, Püntener & Billon-Bruyat (2014), except for NMS 9144 and NMS 37251; NMS 22286-22302 (material from the Solothurn St Niklaus quarry); MJSN BSY008-905 (Figs. 8A–8F); OUMNH J.66966 (Pérez-García, in press; see Discussion).

Diagnosis. Differing from Th. bruntrutana in: proportionally wider nuchal with no anterolateral thickening on ventral surface; narrower vertebral scales; proportionally smaller plastron; narrower and more inclined xiphiplastron; smaller angle between scapular process and acromion process. Differing from Th. marina in: narrower vertebrals scales; less pronounced lateral plastral fontanelles.

Thalassemys bruntrutana sp. nov.

urn:lsid:zoobank.org:act:E3FB882C-BD2B-4C6F-84EE-365033729E74

Figs. 3A–3B, 4A–4B, 5A–5C, 6A–6B, and 7A–7B

Figure 4 Nuchal of Thalassemys bruntrutana.

Specimen MJSN SCR011-87 (Kimmeridgian, Porrentruy, Switzerland). (A) dorsal view; (B) posterior view with the visceral side upward, showing the strong anterolateral thickening of the nuchal.

Figure 5 Peripherals of Thalassemys bruntrutana.

Specimen MJSN SCR011-87 (Kimmeridgian, Porrentruy, Switzerland). (A) Group of peripherals in dorsal view; (B) two peripherals of the bridge area in ventral view; (C) peripheral of the bridge area in lateral view.

Figure 6 Plastra of Thalassemys.

(A, B) Thalassemys bruntrutana, specimen MJSN SCR011-87 (Kimmeridgian, Porrentruy, Switzerland); (C, D) Thalassemys hugii, specimen NMS 8595–8609 (Kimmeridgian, Solothurn, Switzerland). Line width indicates natural borders (thick lines), bone sutures (medium lines), and fractures (thin lines); double lines indicate scale sulci. Abbreviations: cpf, central plastral fontanelle; lpf, lateral plastral fontanelle; hyo, hyoplastron; hypo, hypoplastron; xi, xiphiplastron.

Figure 7 Scapulae of Thalassemys.

Lateral (A) and medial (B) view of the left scapula of Thalassemys bruntrutana, specimen MJSN SCR011-87 (Kimmeridgian, Porrentruy, Switzerland); Lateral (C) and medial (D) view of the left scapula of Thalassemys hugii, specimen NMS 8595–8609 (Kimmeridgian, Solothurn, Switzerland).

Etymology. The species name refers to the German form of the name Porrentruy (Pruntrut), possibly derived from Bruntrutum (meaning abundant springs) (Schweizerische Bundeskanzlei, 2015). This name is also a homage to Jules Thurmann (1804–1855), a renowned local geologist and botanist, for its fundamental paleontological work Lethea bruntrutana that was published after his death (Thurmann & Etallon, 1861–1864).

Holotype. MJSN SCR011-87, an almost complete and articulated carapace (peripherals and nuchal disarticulated), disarticulated plastron, and left scapula.

Type horizon and locality. Lower Virgula Marls, Chevenez Member, Reuchenette Formation (Eudoxus ammonite zone, upper Kimmeridgian, Late Jurassic), vicinity of Porrentruy (Courtedoux village), Canton of Jura, Switzerland (Comment et al., in press).

Occurrence. Kimmeridgian of Switzerland (Porrentruy and Solothurn) and England (Isle of Purbeck).

Referred specimens. NMS 9144 and NMS 37251; NHMUK R8699.

Diagnosis. Differing from Th. hugii in: more elongated nuchal (in anteroposterior direction) that strongly thickens anterolaterally on the ventral surface; wider vertebrals scales; proportionally longer plastron; broader and less inclined xiphiplastron; wider angle between scapular process and acromion process. Differing from Th. marina in: wider vertebrals scales; less pronounced lateral plastral fontanelles.

MJSN SCR011-87 (Thalassemys Bruntrutana)

General preservation

MJSN SCR011-87 consists of an almost complete carapace and plastron associated with the left scapula. The neural series is almost complete and still in articulation with the costals. The nuchal and most of the peripherals are preserved as disarticulated elements. The remaining peripherals, the suprapygals and the pygal are missing. The different elements of the plastron are disarticulated. Epiplastra and entoplastron are missing. The uneven and fractured surface is yellowy-white in color with orange-to-reddish patches (iron mineralizations, mainly on the carapace) and black spots (manganese), the latter occasionally forming tiny dendrites along fractures. There are several traces of serpulids, mainly on the surface of the detached peripherals. Being broad but shallow, the scale sulci are not always easy to discern. Distinct linear striations extend perpendicular to sutures (mainly in anteroposterior direction), as observed in Th. hugii and Th. marina (Anquetin, Püntener & Billon-Bruyat, 2014). These striations are the most striking at the hyo-hypoplastron suture.

Carapace

As preserved, the (sub-complete) carapace measures 477 mm in length along the midline and 531 mm on the maximal width (at the level of costals 3–4). These dimensions are comparable to that of the lectotype of Th. hugii (NMS 8595–8609; Fig. 3). NMS 8595–8609 has a preserved length of about 630 mm (Bräm, 1965). However, without the nuchal and the two suprapygals (elements that are disarticulated or missing in MJSN SCR011-87), it reaches a length of 470 mm. With 530 mm (Bräm, 1965), NMS 8595–8609 has furthermore almost the same preserved width as MJSN SCR011-87 (both specimens missing the peripherals on either side of the carapace). Individual bone measurements (neurals, costals) confirm that MJSN SCR011-87 and NMS 8595–8609 are approximately of the same size (Table S1). MJSN SCR011-87 is clearly larger than the holotype of Th. marina (SMNS 10817). However, the ontogenetic stage of SMNS 10817 is unknown.

The shell is unusually flat. This can be partially explained by taphonomic compaction, which notably flattened the carapace medially along the neurals. However, there are no important openings between the costals, which suggests that the compaction was only moderate. The original carapace vaulting must have been similar to that of Th. hugii (NMS 8595–8609), which is less affected by taphonomic compaction and therefore slightly more domed. Compared to Th. bruntrutana and Th. hugii, the carapaces of plesiochelyids (e.g., Plesiochelys etalloni or Tropidemys langii Rütimeyer, 1873) are clearly more domed.

As preserved, the carapace is roundish in outline, but the missing rim elements (especially the suprapygals and pygal) prevent an accurate reconstruction of the original outline (Figs. 3A and 3B). Bräm (1965) reconstructed the carapacial outline of Th. hugii as slightly heart-shaped, but most of the specimens he had at hand were either too incomplete or missing their peripherals. In contrast, based on undescribed material from Solothurn, Lapparent de Broin, Lange-Badré & Dutrieux (1996) suggested that the carapace of Th. hugii had an oval outline. However, the exact carapacial outline of Th. hugii and Th. marina remains uncertain.

Nuchal

The exact outline of the disarticulated nuchal cannot be reconstructed (Fig. 4A). It is roughly rectangular and proportionally more elongated than the nuchal of Th. hugii (e.g., NMS 8595–8609; Fig. 3D). There is no nuchal notch. The uneven notch on the anterior border of the nuchal is clearly a fracture. A wide embayment on the posterior border of the nuchal once held the first neural. The nuchal bone is thickening strongly anterolaterally on the visceral side (Fig. 4B). This distinctive feature is absent in Th. hugii (see Discussion).

Neurals

There are eight neurals (Figs. 3A and 3B). The first one misses its anterior part, but the wide embayment on the posterior border of the nuchal (see above) indicates a strong widening of the first neural towards the anterior. The following neurals are roughly hexagonal in outline with the shorter sides facing anteriorly (except for neural 8). Neurals 2 and 3 are twice as long as wide. Neurals 4–8 are then successively shorter in proportion, neural 8 being clearly wider than long. The bone posterior to neural 8 is herein interpreted as the ‘intermediate’ element, as described in Th. hugii and several plesiochelyids (Anquetin, Püntener & Billon-Bruyat, 2014).

Costals

Costals 2–5 are completely preserved on each side of the carapace (Figs. 3A and 3B). In contrast, costals 1 have incomplete anterior borders, whereas costals 6–8 lack their lateral parts. The posterior border is convex posteriorly in costals 1 and 2, but concave posteriorly in costals 3–8. As an individual variation, the distal margin of the left costal 2 is clearly longer than that of its right counterpart. The lateral borders of costals 2–5 as well as the posterolateral border of the first costal do not show any sign of suture, indicating a cartilaginous contact with the peripherals (possibly with minor costo-peripheral fontanelles). Although damaged by abrasion, the preserved anterolateral border of the right costal 1 suggests a sutural connection with peripherals 1 and 2. In Th. hugii, the first three peripherals are sutured to the first and second costals, but the condition for other peripherals remains uncertain (NMS 8595–8609; Figs. 3C and 3D).

Peripherals

Among the disarticulated bone elements, seventeen peripherals or parts of peripherals have been identified (Fig. 5). However, their precise position in the carapace cannot be evaluated, mainly due to the poor preservation and the absence of sutural contacts between most peripherals and costals. Some of these peripherals are still partly articulated with one another (Figs. 5A and 5B). The largest peripheral of the bridge area reaches 100 mm in length and 80 mm in width.

Scales of the carapace

As often, scale sulci are difficult to discern on the nuchal bone. One cervical scale can be discerned on the anteromedial part of the nuchal (Fig. 4A). It is about 30 mm long and 50 mm wide. However, the poor surface preservation impedes a conclusion about the actual number of cervicals. Anquetin, Püntener & Billon-Bruyat (2014) suggested that three cervicals may have been present in Th. hugii (based on NMS 8595–8609).

Five vertebrals are present (Fig. 3B). The first vertebral is wider than the nuchal and would have reached the (missing) first peripherals anterolaterally. Vertebrals 2 and 3 are hexagonal in shape with posterolateral margins shorter than anterolateral ones. This can also be observed in Th. marina (SMNS 10817). In Th. hugii, the anterolateral and posterolateral sides tend to be of similar length (e.g., NMS 8595–8609, NMS 8555 and NMS 8997), but some variation may exist for this character (e.g., OUMNH J.66966). In Th. bruntrutana, vertebrals 2 and 3 are about twice as wide as long and cover about half of the costal width. They are proportionally slightly wider than in Th. marina and significantly wider than in Th. hugii, where they cover only about one quarter to one third of the costal width (e.g., NMS 8595–8609, NMS 8612–8627 and NMS 8733). Vertebral 4 is the longest in the series. Its anterolateral sides are mostly straight. The outline of vertebral 5 is not preserved. The intervertebral scale sulci are concave posteriorly and cross neurals 1, 3, 5, and the ‘intermediate’ element.

Pleurals 1–3 can partially be discerned (Fig. 3B). Although their lateral borders are missing, it is evident that they are narrower than vertebrals 2–4. The interpleural 1/2 and 2/3 sulci are situated on the posterior parts of costals 2 and 4, running parallel to the intercostal sutures. Pleurals 4 are not preserved.

Plastron

The hyo-, hypo-, and xiphiplastra are preserved and mostly complete (Figs. 6A and 6B), but the epiplastra and entoplastron are missing. The lateral parts of the hyo- and hypoplastra now lie within the same plane as the rest of the plastron, due to taphonomic compaction. As preserved, the plastron is as wide as long (maximal preserved length = 510 mm; maximal preserved width = 510 mm). Thalassemys bruntrutana has a proportionally longer plastron than Th. hugii. Despite the almost identical carapace size (see above), the plastron of MJSN SCR011-87 is measurably longer than the one of NMS 8595–8609 (Figs. 6C and 6D), which has a maximal preserved length of 430 mm. The length of the bridge measured between the axillary and inguinal notches is also more important in Th. bruntrutana (225 mm in MJSN SCR011-87) than in Th. hugii (190 mm in NMS 8595–8609).

Hyo- and hypoplastra

The hyoplastra are wider than long (Figs. 6A and 6B). A several centimeters wide, obconical notch that once hold the missing entoplastron separates the hyoplastra anteromedially, whereas an elongated central plastral fontanelle separates them posteromedially. In between, on a length of about 50 mm, the hyoplastra meet along an undulating contact. Anteriorly, there is no evidence of sutural contact with the epiplastra and entoplastron.

The hypoplastra are more than twice as wide as long (Figs. 6A and 6B). Separated anteromedially by the central plastral fontanelle, they meet on the posterior 65 mm of their length along an undulating contact. Only about one third of the central plastral fontanelle lies between the hypoplastra, where it is squarish to roundish in shape, in contrast to the longer, rather oval part between the hyoplastra. The lateral margins of the hyoplastra are severely damaged, but the hypoplastra show clear evidence for lateral fontanelles on their anterolateral borders (Figs. 6A and 6B). Based on NMS 8595–8609, Bräm (1965) reconstructed the plastron of Th. hugii without lateral fontanelles, but with an extensive central fontanelle that completely separates the hyo- and hypoplastra. In this reconstruction, only the posterior parts of the xiphiplastra are connected medially. However, the preservation of this specimen does not allow to conclude either to the absence of a lateral plastral fontanelle, or to the size and shape of the central plastral fontanelle (Figs. 6C and 6D). Isolated elements from Solothurn (e.g., NMS 22325) clearly indicate that Th. hugii indeed possesses a lateral plastral fontanelle and an interdigitating contact between the hyoplastra (Anquetin, Püntener & Billon-Bruyat, 2014), refuting Bräm’s reconstruction. We have detected new plastral material of Th. hugii from the St Niklaus quarry in the Solothurn collections (NMS 22286, NMS 22287, and NMS 22296; possibly from the same individual as NMS 22325) that confirm the observations of Anquetin, Püntener & Billon-Bruyat (2014), but it remains uncertain whether the hypoplastra met medially in Th. hugii, as in Th. bruntrutana and Th. marina. The lateral plastral fontanelles of Th. marina are clearly more pronounced than in Th. bruntrutana and Th. hugii.

Xiphiplastra

The xiphiplastra, which are still connected to the hypoplastra by suture, are long elements (about twice as long as wide) that are narrowing strongly posteriorly (Figs. 6A and 6B). They are clearly separated by a fontanelle anteromedially and by a small anal notch posteromedially. It is unclear whether the xiphiplastra actually met one another medially. The xiphiplastra are not as strongly inclined (in relation to the anteroposterior axis of the plastron) as in Th. hugii (e.g., NMS 8595–8609; Figs. 6C and 6D). Compared to Th. hugii, the xiphiplastra are also broader anteriorly at the contact to the hypoplastra.

Scales of the plastron

Scale sulci are only partially preserved (Fig. 6B). They show no important differences from Th. hugii (e.g., NMS 8595–8609; Fig. 6D). The limit between humeral and pectoral scales lies slightly anterior to the level of the deepest point of the axillary notches. The pectorals are slightly shorter than the preserved parts of the humerals. The hyo-hypoplastral suture divides the abdominal scale in about two equally sized parts. The femoral-anal sulcus is not preserved. Of the inframarginal scutes, only the medial borders are partly preserved.

Scapula

Only the left scapula is preserved (Figs. 7A and 7B). The glenoid fossa is only poorly preserved and partly filled with sediment. The dorsally projecting scapular process is complete. It measures 140 mm from the dorsal end to the notch between the acromion process and the coracoid. It is only slightly longer than the scapular process of NMS 8595–8609 (Th. hugii; 135 mm; Figs. 7C and 7D). The scapular process forms an angle of 130° with the acromion process. Due to minor post-mortem deformation, the scapular process and acromion process do not lie exactly in the same plane, which might have a slight influence on the measured angle. In Th. hugii, this angle is always smaller than in Th. bruntrutana (Table 1).

MJSN BSY008-905 (Thalassemys Hugii)

Carapace

The carapace of specimen MJSN BSY008-905 is represented by the nuchal, neurals 3–5, left costals 2–3, right costal 5, right peripherals 1–2, and nine other peripherals or parts of peripherals (Figs. 8A and 8B). All elements of the carapace are disarticulated. The bone surface is brownish-gray with distinct linear striations perpendicular to sutures. This specimen is smaller than the lectotype of Th. hugii (about 85% of the size of NMS 8595–8609 based on individual bone measurements).

Figure 8 MJSN BSY008-905, Thalassemys hugii (Kimmeridgian, Porrentruy, Switzerland).

(A, B) carapace; (C, D) plastron; lateral (E) and medial (F) view of the (left?) scapula. Line width indicates natural borders (thick lines), bone sutures (medium lines), and fractures (thin lines); double lines indicate scale sulci. Abbreviations: co, costal; n, neural; nu, nuchal; v, vertebral scale; hyo, hyoplastron; hypo, hypoplastron; xi, xiphiplastron.

Nuchal

The nuchal of MJSN BSY008-905 (Figs. 8A and 8B) is roughly trapezoidal in outline and about twice as wide as long. Anteriorly, there is a broad and very shallow nuchal notch. The ventral surface of the nuchal is flat and lacks anterolateral thickenings. These are characteristics of Th. hugii and clearly exclude MJSN BSY008-905 from Th. bruntrutana. A wide embayment on the posterior border of the nuchal once held the first neural.

The anterolateral borders of the nuchal are emarginated in order to hold an anteromedial projection of the first peripherals. Such emarginations are usually absent in Th. hugii, but they occur in the specimen OUMNH J.66966 from the Kimmeridge Clay (Pérez-García, in press; see Discussion). Püntener et al. (2014) reported a comparable morphology in one specimen of Tropidemys langii from Porrentruy, although in this specimen small supernumerary bones articulate with the nuchal in this area. This morphology is therefore interpreted as an intraspecific variation.

Neurals

The three preserved neurals are hexagonal in outline with shorter sides facing anteriorly (Figs. 8A and 8B). The neural length decreases from neurals 3 to 5, but all neurals remain clearly longer than wide.

Costals

The lateral border of the second costal is rounded and smooth on the dorsal edge, but shows a sutural contact on the visceral edge, suggesting the transition from a cartilaginous to a sutural contact. Costals 3 and 5 do not show any sign of suture with the peripherals. Here, the contact to the peripherals was fully cartilaginous and minor costo-peripheral fontanelles may have been present. Costal 5 still possesses a laterally jutting rib (Figs. 8A and 8B). The closure of costo-peripheral fontanelles is slightly less advanced than in the only sub-complete adult specimen of Th. hugii (NMS 8595–8609; see Discussion).

Peripherals

Out of the eleven preserved peripherals, the precise position of only two could be identified with certainty: the right peripherals 1 and 2 (Figs. 8A and 8B). The posterior borders of peripherals 1–2 show a sutural contact to the anterior border of the missing first costal. The anteromedial corner of peripheral 1 is reaching out to fit in the anterolateral emargination of the nuchal. The second peripheral is squarish and contrasts with the elongated second peripheral in the lectotype of Th. hugii (NMS 8595–8609). However, the shape of peripherals is known to be variable in other closely related taxa, for example in P. etalloni (Anquetin, Püntener & Billon-Bruyat, 2014).

Scales of the carapace

Cervical scales are only weakly expressed. There is probably a medial cervical scale that is about 15 mm long and 35 mm wide (Fig. 8B). To its left, a narrower lateral cervical scale may have been present. However, no trace of a lateral cervical scale is preserved on the right side. Hence, the actual number of cervicals is not conclusive in this specimen. The second vertebral covers about one quarter to one third of the width of costal 3 and the third vertebral covers slightly more than one third of the width of costal 5 (Fig. 8B). This is congruent with other specimens referred to Th. hugii, but clearly narrower than in Th. bruntrutana (where vertebral scales cover about half of the costal width) and Th. marina. There is no trace of the intervertebral 3/4 scale sulcus on costal 5, nor on neural 5, suggesting a shift of this sulcus onto costal 6 and neural 6 respectively. Such a shift is known to occur occasionally in other taxa, for example in Tr. langii (Püntener et al., 2014).

Plastron

Of the plastron, only parts of the left hyoplastron, right hypoplastron, and right xiphiplastron are preserved (Figs. 8C and 8D). The bridge length is estimated to be around 170 mm. As most borders are broken, the presence of neither lateral nor central fontanelles can be confirmed in this specimen. As in other specimens referred to Th. hugii, the lateral border of the xiphiplastron is strongly inclined relative to the anteroposterior axis of the plastron, which clearly contrasts with the condition in Th. bruntrutana.

Scapula

A small portion of the (left?) scapula is preserved (Figs. 8E and 8F). The distal part of the scapular and acromion processes is missing, as well as the glenoid. The angle between the scapular and acromion processes is about 116°. This falls within the range measured for Th. hugii and contrasts with the larger angle observed in Th. bruntrutana (Table 1).

Discussion

Alpha taxonomy of Thalassemys

In the present study, three species of Thalassemys are considered valid: Th. hugii, Th. marina, and the new species Th. bruntrutana. Thalassemys hugii, the type species, is based on a relatively complete shell associated with some postcranial elements (NMS 8595–8609) from the late Kimmeridgian of Solothurn, Switzerland (Rütimeyer, 1873; Bräm, 1965). The validity of this species has never been questioned. Thalassemys marina is based on a partial carapace and plastron (SMNS 10817) from the Tithonian of Schnaitheim, Germany (Fraas, 1903). Based on the presence of a lateral plastral fontanelle, Bräm (1965) and Maisch (2001) referred this species to the genus Eurysternum. However, it was later demonstrated that a lateral plastral fontanelle was also present in Th. hugii (Anquetin, Püntener & Billon-Bruyat, 2014). Joyce (2003) tentatively synonymized Thalassemys marina with Palaeomedusa testa Meyer, 1860 based on the purported presence of supernumerary pleural scales in SMNS 10817, but the specimen is reconstructed in this area and this assertion does not withstand direct observation. Since SMNS 10817 exhibits significant differences with specimens referred to Th. hugii and Th. bruntrutana (see below), Th. marina must be considered a valid species.

The nuchal of Th. bruntrutana is remarkable in many aspects. It is significantly more elongated than that of Th. hugii and it is characterized by the presence of a strong anterolateral thickening of the ventral surface on both sides. Such morphology has never been observed in other Late Jurassic coastal marine turtles from Europe. However, a similar condition has been described in the lindholmemydid freshwater turtle Amuremys planicostata (Riabinin, 1930) from the Late Cretaceous of Russia, for which it is considered a diagnostic feature (Danilov et al., 2002). The nuchal of Th. marina is not known.

The width of vertebral scales is another distinctive feature between species of the genus Thalassemys. The vertebral scales of Th. hugii are narrow (covering 1/4 to 1/3 of the costal bones), whereas the vertebral scales of Th. bruntrutana are distinctly wider (covering about 1/2 of the costals). The vertebral scales of Th. marina are somewhat intermediate in width between those of Th. hugii and Th. bruntrutana. The width of vertebral scales is known to change during ontogeny, usually decreasing from juveniles to adults (e.g., Joyce, 2007). However, it is noteworthy that juvenile specimens of Th. hugii (e.g., NMS 8612–8627, NMS 8733, and NMS 8997) have the same narrow vertebral scales as the adults (e.g., NMS 8595–8609 and NMS 8555). It must also be considered that the type specimens of Th. hugii and Th. bruntrutana are of similar size. Therefore, ontogeny cannot explain the clearly different width of vertebral scales in the two taxa. The case of Th. marina is more complicated since this species is known only by a single individual that is about 65% the size of the type specimens of the two other species. However, two arguments allow the hypothesis that the holotype of Th. marina would be a juvenile of either Th. hugii or Th. bruntrutana to be rejected. First, as aforementioned, juveniles of Th. hugii have narrow vertebral scales. Second, given the general tendency toward the reduction of the vertebral scale width during ontogeny, juveniles of Th. bruntrutana would be expected to have vertebral scales that are wider than, or at least as wide as, those of the adults.

At comparable size, the suturing of costals 1 and 2 with adjoining peripherals is more advanced in Th. hugii than in Th. bruntrutana. For example, in the lectotype of Th. hugii (NMS 8595–8609) peripherals 1–3 are sutured to costals 1 and 2 at least up to the anterior half of peripheral 3. In the holotype of Th. bruntrutana (MJSN SCR011-87; a similarly-sized specimen), only peripherals 1 and 2 are sutured with costal 1, whereas the posterolateral border of costal 1 and costals 2–5 lack sutural contacts with the peripherals (see above). MJSN BSY008-905, a specimen we refer to Th. hugii, confirms this difference in the timing of costo-peripheral suturing between the two species. An incipient sutural contact between costal 2 and peripheral 3 is present in this specimen, although its size is only about 85% that of the holotype of Th. bruntrutana. This area is not preserved in Th. marina, which prevents comparison.

Lateral plastral fontanelles are present in all three species of Thalassemys. In Th. hugii and Th. bruntrutana, the lateral plastral fontanelle is relatively narrow, even in juvenile specimens. This probably explains why the presence of these fontanelles long went unnoticed in Th. hugii (see Anquetin, Püntener & Billon-Bruyat, 2014). In contrast, the lateral plastral fontanelle is significantly broader in Th. marina.

Thalassemys hugii is remarkable in the strong inclination of the lateral border of its xiphiplastron relative to the anteroposterior axis of the plastron (Table 2). In Th. bruntrutana and most other turtles, the lateral border of the xiphiplastron is significantly less inclined in relation to the anteroposterior axis. The xiphiplastron is unknown in Th. marina. Two specimens from Solothurn previously referred to Th. hugii (NMS 9144 and NMS 37251; see Anquetin, Püntener & Billon-Bruyat, 2014) are herein tentatively referred to Th. bruntrutana based on the morphology of their xiphiplastra (Table 2).

Table 2 The angle between the anteroposterior axis of the plastron and the lateral border of the xiphiplastron in specimens of Thalassemys.

Specimen	Species	Xiphiplastral angle	
NMS 8595–8609	Th. hugii	45°	
MJSN BSY008-905	Th. hugii	49°	
OUMNH J.66966	Th. hugii a	48°	
NMS 9144	Th. bruntrutana b	30°	
NMS 37251	Th. bruntrutana b	29°	
MJSN SCR011-87	Th. bruntrutana	30°	
Notes.

a cf. Pérez-García (in press).

b Thalassemys hugii in Anquetin, Püntener & Billon-Bruyat (2014).

As noted by Bräm (1965), the angle between the scapular and acromion process of the scapula is more open in Thalassemys than in the plesiochelyid Plesiochelys etalloni. Within Thalassemys, the scapular angle ranges from 113° to 122° in specimens referred to Th. hugii (including MJSN BSY008-905), whereas this angle reaches 130° in the holotype of Th. bruntrutana (Table 1). Although the scapular angle of Th. marina is unknown, it appears that this measurement can be used to distinguish Th. hugii from Th. bruntrutana.

The potential effect of ontogeny and sexual dimorphism on our perception of the alpha taxonomy of Thalassemys must be considered. As discussed above, ontogeny can be easily dismissed to explain the differences between the three identified species. Sexual dimorphism was considered seriously notably for Th. hugii and Th. bruntrutana, which are contemporaneous species sometimes occurring in the same localities (see Paleobiogeographic considerations). In recent turtles, sexual dimorphism is primarily expressed in a difference in shell size between adult males and females (Berry & Shine, 1980; Pritchard, 2008). Furthermore, adult males often develop a concave plastron (in terrestrial species) and a shorter and wider anal notch (Pritchard, 2008). In terms of fossil turtles, reports of sexual dimorphism are scarce and relate for instance to the shape of the anal notch, to the size of the central plastral fontanelle (Cadena, Jaramillo & Bloch, 2013), to plastral kinesis, and to tail length (Joyce et al., 2012). All these differences are directly linked to sexual selection (shell size), copulation (concave plastron, plastral fontanelle size, shape of anal notch, tail length) or oviposition (shape of anal notch, plastral kinesis) (Berry & Shine, 1980; Pritchard, 2008; Joyce et al., 2012; Cadena, Jaramillo & Bloch, 2013). In contrast, we are unable to link any observed anatomical difference between Th. hugii and Th. bruntrutana (e.g., the vertebral width or the nuchal shape) to reproductive behavior. Therefore, we consider these differences as specific and interpret Th. hugii and Th. bruntrutana as two closely related species.

Thalassemys from the Kimmeridge Clay Formation

Recently, Pérez-García (2015) discussed a relatively complete, but strongly flattened carapace of Thalassemys with associated postcranial remains from the Kimmeridge Clay Formation (late Kimmeridgian) of Egmont Bight, Isle of Purbeck, Dorset, England (NHMUK R8699). Based on the presence of linear striations perpendicular to sutures and the characteristic outline of the vertebral scales, we agree that this specimen belongs to Thalassemys. Pérez-García (2015) noted some differences between this specimen and the lectotype of Th. hugii (smaller size, wider vertebral scales, and more developed costo-peripheral fontanelles) and safely concluded that no specific determination was possible for this specimen at that time. NHMUK R8699 and the holotype of Th. bruntrutana (MJSN SCR011-87) have several features in common. Although NHMUK R8699 is only about 65% the size of MJSN SCR011-87, both specimens have vertebral scales of the same proportions (about twice as wide as long) and shape (clearly longer anterolateral sides). As discussed above, the vertebral width likely represents a specific character within Thalassemys, in that case uniting NHMUK R8699 with Th. bruntrutana and distinguishing it from Th. hugii and Th. marina. The scapular angle measured on NHMUK R8699 is relatively small (about 103°; not 115° as incorrectly noted by Pérez-García, 2015), but it should be noted that this specimen has been severely flattened during fossilization: all bones and shell plates are flat and thin. This measured angle probably does not reflect the original scapular angle in this specimen. As discussed above, we can also observe a very slight deformation of the scapula of MJSN SCR011-87. However, NHMUK R8699 has clearly been more flattened during fossilization. There is almost no volume in the scapula, indicating a really strong deformation. Keeping in mind this uncertainty and the fact that several important parts of the shell (nuchal, plastron) are missing, we tentatively refer NHMUK R8699 to Th. bruntrutana and thereby report the presence of this species in the Kimmeridgian of southern England.

In 1992, Richard Wilkins, an amateur geologist, discovered the partial shell and some postcranial elements of a large turtle in the Kimmeridge Clay of Abington, Oxfordshire, England. He tentatively identified the specimen as a thalassemydid and donated it to the Oxford University Museum, where it still resides today (OUMNH J.66966). This specimen was later studied as part of a Master thesis and believed to be an indeterminate pleurodire (Harrison, 1999). However, this shell truly belongs to a thalassemydid and was recently referred to Th. hugii (Pérez-García, in press). Based on the morphology of the nuchal, width of the vertebral scales, and inclination of the xiphiplastron, we agree with this attribution.

Paleobiogeographic considerations

The paleobiogeographic distribution of Late Jurassic coastal marine turtles in Europe is largely unexplored. This is mainly the result of a poor understanding of the alpha taxonomy of these turtles. A global revision of these groups at the European scale is needed. Until recently, Thalassemys hugii was confidently identified only in Solothurn, Switzerland (Rütimeyer, 1873; Bräm, 1965), whereas Th. marina was known by a single specimen from Schnaitheim, Germany (Fraas, 1903). Recently, Pérez-García (in press) referred a specimen from the Kimmeridge Clay of England to Th. hugii and proposed that a second, unidentified species of Thalassemys was also present in the same formation.

In the present study, we identified a new thalassemydid from Porrentruy, Switzerland (Thalassemys bruntrutana), and tentatively proposed that this species was also present in Solothurn and the Isle of Purbeck (the unidentified specimen of Pérez-García, in press). We also described a new specimen from Porrentruy that can be confidently identified as Th. hugii. Our results therefore show that both Th. hugii and Th. bruntrutana are present in the Kimmeridgian of the Swiss Jura Mountains (Solothurn and Porrentruy) and of southern England (Isle of Purbeck and Abington; Fig. 9). This confirms the results of Pérez-García (in press). Thalassemys hugii and Thalassemys bruntrutana represent the first Late Jurassic coastal marine turtles that are demonstrated to have a Western European paleobiogeographic distribution. Such a result is not surprising since some thalattosuchian crocodylomorphs that inhabit the same environments also have a Western European distribution at that time, e.g., the large teleosaurid Machimosaurus hugii Meyer, 1837 (Young et al., 2014; Martin, Vincent & Falconnet, in press). We expect that future studies will also extend the paleobiogeographic repartition of other Late Jurassic coastal marine turtles from Europe.

Figure 9 Paleobiogeographic distribution of Thalassemys.

Localities with Thalassemys species from the Kimmeridgian (black stars) and Tithonian (white star) on a Late Jurassic paleogeographic map of Western Europe. Map from Ron Blakey, Colorado Plateau Geosystems, Arizona, USA (http://cpgeosystems.com/paleomaps.html).

Conclusions

Thalassemys was hitherto mainly based on a relatively complete shell of Th. hugii from the Kimmeridgian of Solothurn (Switzerland). Undetermined thalassemydid material was also reported from different Western European countries (e.g., Germany, England, and France), but poor preservation limited the value of these specimens. The new material from the Kimmeridgian of Porrentruy (Switzerland) offers new insights into the anatomy, taxonomy and paleobiogeographic distribution of Thalassemys. The new species Th. bruntrutana shows important anatomical differences to Th. hugii that cannot be explained by ontogenetic variation or sexual dimorphism. Both species are simultaneously present in the Kimmeridgian of Solothurn and Porrentruy (two localities from the Swiss Jura Mountains of slightly different ages) as well as in the Kimmeridgian of southern England. These results confirm that at least some Late Jurassic coastal marine turtles had a Western European paleobiogeographic distribution. Thalassemys hugii and Th. bruntrutana are currently not identified in the German fossil record, but the undetermined material of Thalassemys from the Kimmeridgian of Oker (northern Germany; Marinheiro & Mateus, 2011) should be analyzed in the light of the Porrentruy material. So far, Th. marina from the Tithonian of Schnaitheim (southern Germany) remains the only valid species of Thalassemys in Germany.

Supplemental Information

Table S1 Individual bone measurements in MJSN SCR011-87 (Th. bruntrutana) and NMS 8595–8609 (Th. hugii)

Click here for additional data file.

We thank Loïc Bocat (excavation), Renaud Roch (preparation), Olivier Noaillon and Bernard Migy (photographs), Pierre Widder (scientific drawings), Apolline Lefort (discussion on stratigraphy) and the whole Paleontology A16 team. Further thanks go to Silvan Thüring of the Naturmuseum Solothurn for providing access to the lectotype of T. hugii, and to Adán Pérez-García for discussions on Thalassemys. Editor Mathew Wedel and the reviewers Walter Joyce and Márton Rabi provided very helpful comments.

Institutional abbreviations

MJSN JURASSICA Museum (formerly Musée jurassien des sciences naturelles), Porrentruy, Switzerland

NHMUK Natural History Museum, London, UK

NMS Naturmuseum Solothurn, Switzerland

OUMNH Oxford University Museum of Natural History, Oxford, UK

SMNS Staatliches Museum für Naturkunde Stuttgart, Germany

Locality abbreviations

BSY Courtedoux—Bois de Sylleux

SCR Courtedoux—Sur Combe Ronde.

Additional Information and Declarations

Competing Interests

Author Contributions

New Species Registration

Jérémy Anquetin is an Academic Editor for PeerJ.

Christian Püntener and Jérémy Anquetin conceived and designed the experiments, performed the experiments, analyzed the data, wrote the paper, prepared figures and/or tables, reviewed drafts of the paper.

Jean-Paul Billon-Bruyat analyzed the data, wrote the paper, reviewed drafts of the paper.

The following information was supplied regarding the registration of a newly described species:

Thalassemys bruntrutana sp. nov.

urn:lsid:zoobank.org:act:E3FB882C-BD2B-4C6F-84EE-365033729E74.

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
