# Peer review of "Thalassemys bruntrutana n. sp., a new coastal marine turtle from the Late Jurassic of Porrentruy (Switzerland), and the paleobiogeography of the Thalassemydidae"

_PeerJ, doi:10.7717/peerj.1282_

## Round 0.1 · original submission · Minor Revisions

Both reviewers found the manuscript to be well-written and important and, as you will see, their recommendations for improvement are reasonable and straightforward. Please be diligent in addressing their concerns, and I will look forward to seeing an even tighter version of the manuscript soon.

·

Basic reporting

all fine (see comments below)

Experimental design

all fine (see comments below)

Validity of the findings

all fine (see comments below)

Additional comments

This manuscript is concerned with the description of two new turtle specimens from northwestern Switzerland referable to Thalassemys, the naming of a new species of Thalassemys, and a synthesic review of the available record of Thalassemys in Europe.
All in all this is an extremely clean, well-written paper and I have only little to add. The photographs are sharp, the illustrations clear, all conclusions well backed up by data, and the results are relevant.
I marked up the manuscript with a very small number of comments that may help improve the quality of the manuscript, including some minor improvements to the figures, especially addition of Th. marina to figure 8. The suggested changes, however, are extremely minor and I therefore see no need for an additional round of review.
The authors are welcome to know my identity.
Best regards,

Walter Joyce

·

Basic reporting

No Comments.

Experimental design

No Comments.

Validity of the findings

No Comments.

Additional comments

This is a very thorough paper and an important contribution for the taxonomy and paleobiogeography of Late Jurassic European turtles. Descriptions are very detailed, illustrations are of high quality and I find the interpretation of the available data fully valid.

The authors really should comment on the recently named taxon Jurassichelon orlensis that used to be "Thalassemys" moseri. Do they agree with the new attribution? There was no phylogenetic analysis in that paper that would justify the erection of a new genus. How do the authors relate Jurassichelon orlensis to Thalassemys spp.?

Thalassemydidae is still a monogeneric family. Do we really need such a family to be kept until there are no other genera included in it? I think not and I would suggest abandoning it for the moment. Or isn't Jurassichelon also considered a thalassemydid?

Please use another term instead of "European" distribution since the record is still very incomplete and there are only a few datapoints. The fossil record certainly does not span the entire area of Europe at this point. Moreover, the English and Swiss populations in theory could have been isolated even though it is admittedly unlikely.

I suggest using the nomenclature of Joyce et al. 2004 and Pancryptodira instead of Eucryptodira (at least in the case of the Systematic section).

The term "Crocodylia" is reserved for the crown-group, please use instead crocodilians.

Th. cf. bruntrutana would be a more conservative referral for the English occurrence. I suggest reconsidering the current assignment.

---

## Round 0.2 · Minor Revisions

Thank you for your diligence in responding to the reviewers' suggestions for improvement. I am writing to request just a handful of very minor changes, which will be easy to implement, after which your paper will be acceptable for publication.

First, your comments in the rebuttal letter on the status of "T." moseri, the use of the monogeneric family Thalassemydidae, and the deformation of the scapula are interesting and useful - please incorporate them directly into the manuscript. The comments as they appear in the rebuttal letter are sufficiently cogent and well-written that you could basically copy and paste them into the manuscript with just enough rewording to smooth over the seams.

Second, I agree that your figures are readable as-is, without the taxon names in the figures, but I urge you to consider that your figures may be used by other researchers as reference material and in their own talks and papers (with proper attribution, consistent with PeerJ's open-access CC-BY license). Including the taxon names will improve the reusability of the figures - and hence the citability of your paper. Since there are potentially large downstream benefits to including the taxon names, and essentially zero cost (it should be a five-minute operation in your graphics software), I would prefer you to accept the reviewer's request on that point.

After these changes are made, I see no further impediments to the swift acceptance and publication of your manuscript.

---

## Round 0.3 · accepted · Accept

Thank you for your prompt attention in revising the manuscript. I'm happy to accept it for publication in PeerJ.

You have the option to publish the peer review history alongside the paper. That decision is entirely up to you, and it will have no bearing on how the paper is treated. I think there is much to be gained from publishing the peer reviews - this was a good example of constructive reviews improving an already solid paper. I leave it in your hands.